# DIMENSIONAL REWEIGHTING GRAPH CONVOLUTIONAL NETWORKS

## ABSTRACT

In this paper, we propose a method named **D**imensional **r**eweighting **G**raph **C**onvolutional **N**etworks (DrGCNs), to tackle the problem of variance between dimensional information in the node representations of GCNs. We prove that DrGCNs have the effect of stabilizing the training process by connecting our problem to the theory of the mean field. However, practically, we find that the degrees DrGCNs help vary severely on different datasets. We revisit the problem and develop a new measure $K$ to quantify the effect. This measure guides when we should use dimensional reweighting in GCNs and how much it can help. Moreover, it offers insights to explain the improvement obtained by the proposed DrGCNs. The dimensional reweighting block is light-weighted and highly flexible to be built on most of the GCN variants. Carefully designed experiments, including several fixes on duplicates, information leaks, and wrong labels of the well-known node classification benchmark datasets, demonstrate the superior performances of DrGCNs over the existing state-of-the-art approaches. Significant improvements can also be observed on a large scale industrial dataset.

## 1 INTRODUCTION

Deep neural networks (DNNs) have been widely applied in various fields, including computer vision (He et al., 2016; Hu et al., 2018), natural language processing (Devlin et al., 2019), and speech recognition (Abdel-Hamid et al., 2014), among many others. Graph neural networks (GNNs) is proposed for learning node presentations of networked data (Scarselli et al., 2009), and later be extended to graph convolutional network (GCN) that achieves better performance by capturing topological information of linked graphs (Kipf & Welling, 2017). Since then, GCNs begin to attract board interests. Starting from GraphSAGE (Hamilton et al., 2017) defining the convolutional neural network based graph learning framework as *sampling* and *aggregation*, many follow-up efforts attempt to enhance the sampling or aggregation process via various techniques, such as attention mechanism (Veličković et al., 2018), mix-hop connection (Abu-El-Haija et al., 2019) and adaptive sampling (Huang et al., 2018).

In this paper, we study the node representations in GCNs from the perspective of covariance between dimensions. Suprisingly, applying a dimensional reweighting process to the node representations may be very useful for the improvement of GCNs. As an instance, under our proposed reweighting scheme, the input covariance between dimensions can be reduced by 68% on the Reddit dataset, which is extremely useful since we also find that the number of misclassified cases reduced by 40%, compared with the previous SOTA method.

We propose **D**imensional **r**eweighting **G**raph **C**onvolutional **N**etworks (DrGCNs), in which the input of each layer of the GCN is reweighted by global node representation information. Our discovery is that the experimental performance of GCNs can be greatly improved under this simple reweighting scheme. On the other hand, with the help of mean field theory (Kadanoff, 2009; Yang et al., 2019), this reweighting scheme is also proved to improve the stability of fully connected networks, provding insight to GCNs. To deepen the understanding to which extent the proposed reweighting scheme can help GCNs, we develop a new measure to quantify its effectiveness under different contexts (GCN variants and datasets).

Experimental results verify our theoretical findings ideally that we can achieve predictable improvements on public datasets adopted in the literature over the state-of-the-art GCNs. While studying on

these well-known benchmarks, we notice that two of them (Cora, Citeseer) suffer from duplicates and feature-label information leaks. We fix these problems and offer refined datasets for fair comparisons. To further validate the effectiveness, we deploy the proposed DrGCNs on A* [1] company's recommendation system and clearly demonstrate performance improvements via offline evaluations.

## 2 DRGCNs: DIMENSIONAL REWEIGHTING GRAPH CONVOLUTIONAL NETWORKS

### 2.1 PRELIMINARIES

**Notations.** We focus on undirected graphs $\mathcal{G} = (\mathcal{V}, \mathcal{E}, \mathbf{X})$, where $\mathcal{V} = \{v_i\}$ represents the node set, $\mathcal{E} = \{(v_i, v_j)\}$ indicates the edge set, and $\mathbf{X}$ stands for the node features. For a specific GCN layer, we use $\mathbf{R}^{in} = (\mathbf{r}_1^{in}, ..., \mathbf{r}_n^{in})$ to denote the input node representations and $\mathbf{R}^{out} = (\mathbf{r}_1^{out}, ..., \mathbf{r}_n^{out})$ to symbolize the output representations.[2] For the whole layer-stacked GCN structure, we use $\mathbf{H}^0$ to denote the input node representation of the first layer, and $\mathbf{H}^l(l > 0)$ to signify the output node representation of the $l^{th}$ layer, which is also the output representation of the $(l-1)^{th}$ layer. Let $\mathbf{A}$ be the adjacency matrix with $a_{ij} = 1$ when $(v_i, v_j) \in \mathcal{E}$ and $a_{ij} = 0$ otherwise.

**Graph Convolutional Networks (GCNs).** Given the input node set $\mathcal{V}$, the adjacency matrix $\mathbf{A}$, and the input representations $\mathbf{R}^{in}$, a GCN layer uses such information to generate output representations $\mathbf{R}^{out}$:

$$\mathbf{R}^{out} = \sigma(aggregator(\mathbf{R}^{in}, \mathbf{A})), \tag{1}$$

where $\sigma$ is the activation function. Although there exist non-linear aggregators like the LSTM aggregator (Hamilton et al., 2017), in most GCN variants the aggregator is a linear function which can be viewed as a weighted sum of node representations among the neighborhood (Kipf & Welling, 2017; Huang et al., 2018), followed by a matrix multiplication on a refined adjacency matrix $\tilde{\mathbf{A}}$, with a bias added. The procedure can be formulated as follows:

$$\mathbf{R}^{out} = \sigma(\mathbf{W}\mathbf{R}^{in}\tilde{\mathbf{A}} + \mathbf{b}), \tag{2}$$

where $\mathbf{W}$ is the projection matrix and $\mathbf{b}$ denotes the bias vector. Development on GCNs mainly lies in different ways to generate $\tilde{\mathbf{A}}$. GCN proposed some variants including simply taking $\tilde{\mathbf{A}} = \mathbf{A}\mathbf{D}^{-1}$, which is uniform average among neighbors with $\mathbf{D}$ being the diagonal matrix of the degrees, or weighted by degree of each node $\tilde{\mathbf{A}} = \mathbf{D}^{-\frac{1}{2}}\mathbf{A}\mathbf{D}^{-\frac{1}{2}}$, or including self-loops $\tilde{\mathbf{A}} = \mathbf{I} + \mathbf{D}^{-\frac{1}{2}}\mathbf{A}\mathbf{D}^{-\frac{1}{2}}$. Other methods include attention (Veličković et al., 2018), or gated attention (Zhang et al., 2018), or even neural architecture search methods (Gao et al., 2019) to generate $\tilde{\mathbf{A}}$. To improve scalability, some GCN variants contain a sampling procedure, which samples a subset of the neighborhood for aggregation (Chen et al., 2018; Huang et al., 2018). We can set all unsampled edges to 0 in $\tilde{\mathbf{A}}$ in sampling-based GCNs, in this case $\tilde{\mathbf{A}}$ even has some randomness.

### 2.2 MODEL FORMULATION FOR DRGCNs

Given input node representations of a GCN layer $\mathbf{R}^{in}$, the proposed DrGCN tries to learn a dimensional reweighting vector $\mathbf{s} = (s_1, ..., s_d)$, where $s_i$ is an adaptive scalar for each dimension $i$. This reweighting vector $\mathbf{s}$ then helps reweighting each dimension of the node representation $\mathbf{r}_v^{in}$ to $\mathbf{r}_v^{re}$, $v \in \mathcal{V}$, where $\mathbf{r}_v^{re} = \mathbf{r}_v^{in} \circ \mathbf{s}$. Here we use $\circ$ to denote component-wise multiplication, i.e.,

$$r_{v,j}^{re} = s_j r_{v,j}^{in}, \forall 1 \leq j \leq d, \forall v \in \mathcal{V}. \tag{3}$$

We define $\mathbf{S}$ as the diagonal matrix with diagonal entries corresponding to the components of $\mathbf{s}$. Then a DrGCN layer can be formulated as:

$$\mathbf{R}^{out} = \sigma(\mathbf{W}\mathbf{S}\mathbf{R}^{in}\tilde{\mathbf{A}} + \mathbf{b}). \tag{4}$$

Inspired by SENet (Hu et al., 2018), we formulate the learning of the shared dimensional reweighting vector $\mathbf{s}$ in two stages. First we generate a global representation $\mathbf{r}^{in}$, whose value is the expectation of $\mathbf{r}_v^{in}$ on the whole graph. Then we feed $\mathbf{r}^{in}$ into a two-layer neural network structure to generate

---

[1] To preserve anonymity we use A* for the company and dataset name.

[2] For the convenience of our analysis, we use columns of $\mathbf{R}, \mathbf{H}, \mathbf{X}$ instead of rows to represent node representations.

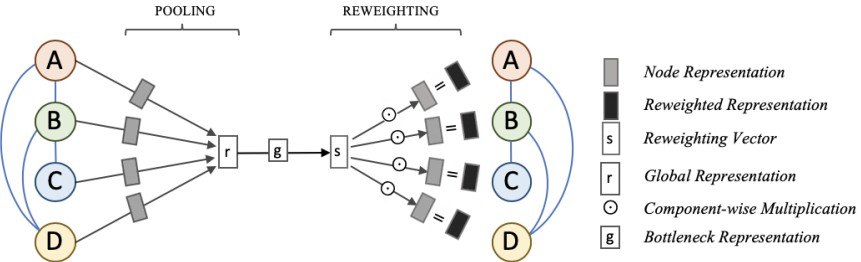

Figure 1: Our proposed dimensional reweighting block (Dr Block) in DrGCNs.

$\mathbf{s}$ of the same dimension size. Equation (5) denotes the procedure to generate $\mathbf{s}$ given node weight $\{w_v | v \in \mathcal{V}, \sum_{v \in \mathcal{V}} w_v = 1\}$ and node representations $\{\mathbf{r}_v^{in} | v \in \mathcal{V}\}$:

$$
\begin{aligned}
\mathbf{r}^{in} &= E[\mathbf{r}_v^{in} | v \in \mathcal{V}] = \sum_{v \in \mathcal{V}} w_v \mathbf{r}_v^{in}, \\
\mathbf{g} &= \sigma_g(\mathbf{W}_g \mathbf{r}^{in} + \mathbf{b}_g), \\
\mathbf{s} &= \sigma_s(\mathbf{W}_s \mathbf{g} + \mathbf{b}_s),
\end{aligned}
\tag{5}
$$

where $\mathbf{g}$ is output of the first layer; $\mathbf{W}_g, \mathbf{b}_g, \mathbf{W}_s$, and $\mathbf{b}_s$ are parameters to be learnt. Figure 1 summarizes the dimensional reweighting block (Dr Block) in DrGCNs.

**Combining With Existing GCN variants.** The proposed Dr Block can be implemented as an independent functional process and easily combined with GCNs. As shown in equation (4), Dr Block only applies on $\mathbf{R}^{in}$ and does not involve in the calculation of $\tilde{\mathbf{A}}, \mathbf{W}$ and $\mathbf{b}$. Hence, the proposed Dr Block can easily be combined with existing sampling or aggregation methods without causing any contradictions. In § 4, we will experimentally test the combination of our Dr Block with different types of sample-and-aggregation GCN methods. Suppose that the input features are $\mathbf{X}$, DrGCNs can be viewed as follows:

$$
\mathbf{H}^l = \sigma_l(\mathbf{W}^l \mathbf{S}^l \mathbf{H}^{l-1} \tilde{\mathbf{A}}^l + \mathbf{b}^l), \forall 1 \leq l \leq k,
\tag{6}
$$

where $\mathbf{H}^0 = \mathbf{X}$ and $\mathbf{H}^k$ being the output representation for a $k$-layer DrGCN:

**Complexity of Dr Block.** Consider a GCN layer with $a$ input and $b$ output channels, $n$ nodes and $e$ edges in a sampled batch, the complexity of a GCN layer is $O(abn+be)$. The proposed Dr block has a complexity of $O(ag)$, where $g$ is the dimension of $\mathbf{g}$. In most cases, we have $g < b$ and $n >> 1$, so we could have $O(ag) = o(abn + be)$, which indicates that Dr block introduces negligible extra computational cost.

## 3 THEORETICAL ANALYSES

In this section, we connect our study to mean field theory (Yang et al., 2019). We theoretically prove that the proposed Dr Block is capable of reducing the learning variance brought by perturbations on the input, making the update more stable in the long run. To deepen the understanding, we further develop a measure to quantify the stability gained by Dr block.

### 3.1 MEAN FIELD APPROXIMATION

(Lee et al., 2018; Yang et al., 2019) employ mean field approximation to analyze fully-connected networks. Following their ideas, we provide theoretical analyses for Dr Blocks on fully-connected networks. GCNs are different from fully-connected networks only in $\tilde{\mathbf{A}}$, and degrade to fully-connected networks when $\tilde{\mathbf{A}} = \mathbf{I}$, our idea is to provide insight to GCNs from the analysis of fully-connected networks. We assume the average of the data is 0 in the following discussions as transformation does not affect the covariance structure. For simplicity, we only consider neural networks with constant width and assume all layers use the same activation function $\phi$. We follow the pre-activation recurrence relation $h_i^l = \mathbf{W}^l \sigma_l(\mathbf{S}^l \circ h_i^{l-1}) + \mathbf{b}^l$ in (Yang et al., 2019) to facilitate the problem. When $S$ being a diagonal matrix with positive entries, and $\phi$ is ReLU activation, $\phi(\mathbf{SH}) = \mathbf{S}\phi(\mathbf{H})$ holds for all $\mathbf{H}$. We can take the pre-activation step as the post-activation step of the previous layer and generalize our analysis to post-activation. So the recursive relation is:

$$
\mathbf{H}^l = \mathbf{W}^l \phi(\mathbf{S}^l \mathbf{H}^{l-1}) + \mathbf{b}^l.
\tag{7}
$$

We apply the mean field approximation to replace the input data by a normal random variable with the same mean and variance and define an estimator $V_\phi$ to characterize the fluctuation of the random variable $\phi(\mathbf{S}^l \mathbf{H})$.

**Define:**
$$V_\phi(\mathbf{S}^l, \mathbf{C}^{l-1}) \triangleq E[\phi(\mathbf{S}^l \mathbf{H})\phi(\mathbf{S}^l \mathbf{H})^T]. \tag{8}$$

In this equation $\mathbf{H} \sim N(0, \mathbf{C}^{l-1})$, $\phi$ is the ReLU activation, and $\mathbf{C}^l$ represents the covariance matrix of $\mathbf{H}^l$, i.e. $\mathbf{C}^l = Cov(\mathbf{H}^l)$. Note that $V_\phi$ does not completely coincide with variance, but can reflect the covariance structure. We call it **covariance matrix** for convenience. With mean field approximation, the covariance matrix can be updated by (note that $\mathbf{b}^l$ is independent with $\mathbf{H}$):
$$\mathbf{C}^l = \mathbf{W}^l V_\phi(\mathbf{S}^l, \mathbf{C}^{l-1})(\mathbf{W}^l)^T + Cov(\mathbf{b}^l). \tag{9}$$

We assume this dynamical system has a BSB1(Block Symmetry Breaking 1 (Yang et al., 2019)) fixed point (where $\mathbf{C}^l = \mathbf{C}^{l-1}$), i.e. a solution having the form $\mathbf{C}^* = q^*((1-c^*)\mathbf{I} + c^*\vec{1}\vec{1}^T)$ of the following equation with respect to $\mathbf{C}$:
$$\mathbf{C} = \mathbf{W}^l V_\phi(\mathbf{S}^l, \mathbf{C})(\mathbf{W}^l)^T + Cov(\mathbf{b}^l). \tag{10}$$

Next we make a reduction of $V_\phi$ so that only the second slot matters. Take $\mathbf{S}^l h$ as a whole, it follows $N(0, \mathbf{S}^l \mathbf{C}^{l-1} \mathbf{S}^l)$ distribution. Note that $\mathbf{S}$ is diagonal so $\mathbf{S} = \mathbf{S}^T$. Thus equation (9) can be written as:
$$\mathbf{C}^l = V_\phi(\mathbf{I}, \mathbf{S}^l \mathbf{C}^{l-1} \mathbf{S}^l) + Cov(\mathbf{b}^l). \tag{11}$$

The derivative of $\mathbf{C}^l$ measures the fluctuation of our updating algorithm when input perturbations exist, hence it characterizes the sensitivity of the algorithm to the data structures and its robustness. We will turn to show that Dr can relieve this sensitivity. We fix the point $\mathbf{C}^l$ where we are taking derivative at. For most common types of activation functions, this recursive map has a fixed point, at which this linearization is most useful. Recall that such a derivative will be a linear map from symmetric matrices to symmetric matrices.

**Define:**
$$J_\phi(\mathbf{C}_1) \triangleq \frac{dV_\phi(\mathbf{I}, \mathbf{C})}{d\mathbf{C}}\Big|_{\mathbf{C}^l}(\mathbf{C}_1). \tag{12}$$

Here $\mathbf{C}_1$ could be intuitively understood as the increment near $\mathbf{C}^l$. We denote by $H_d$ the space of symmetric matrices of size $d \times d$. Using these notations, we prove that:

**Theorem 1.** *There exist diagonal matrices $\mathbf{S}^l$, constants $0 < \gamma^l < 1$ such that, $||J_\phi(\mathbf{S}^l \mathbf{C} \mathbf{S}^l)||_F \leq \gamma^l ||J_\phi(\mathbf{C})||_F \frac{||\mathbf{S}^l||_F}{d}$ for any fixed general $\mathbf{C}$. By general, we mean there exists a Haar measure on the collection of symmetric matrices $H_B$ with respect to which the statement fails has measure zero* [3].

Detailed proofs and explanations are included in the Appendix G,H. For symmetric matrices, with $\lambda_i$ denoting eigenvalues of $\mathbf{A}$, we have:
$$||\mathbf{A}||_F^2 = \sum_{i=1}^d \lambda_i^2. \tag{13}$$

This norm measures the overall magnitude of eigenvalues of the operator. This result demonstrates that our method brings variance reduction and improves the stability of the updating algorithms. To summarize, for any input data, there exists a vector $\mathbf{s}$ that improves the stability of the updating algorithms.

## 3.2 STABILITY MEASURE FOR DRGCNS

In this section we turn to define a quantified measurement of the improvement of the stability of DrGCNs.

**Define:**
$$K \triangleq \frac{\sum_i c_{ii} s_{li}^2 - \frac{1}{d} \sum_{i,j} c_{ij} s_{li} s_{lj}}{(\sum_i c_{ii} - \frac{1}{d} \sum_{i,j} c_{ij}) \times \frac{\sum_i s_{li}^2}{d}}, \tag{14}$$

where $c_{ij}$ is the $(i,j)^{th}$ element of the convariance matrix $\mathbf{C}$. Theorem 2 suggests that K measures the instability of the update. The measure is a relative covariance measure that when $\mathbf{S} = \mathbf{I}$ (without Dr), $K = 1$. This quantity only involves entries of $\mathbf{C}$, and it is homogeneous of degree 0 with respect to these entries and invariant under scalar multiplication on these entries. Being the covariance

---

[3] $|| \cdot ||_F$ is the Frobenius norm, i.e. for a matrix $\mathbf{A} = (A_{ij})$, $||\mathbf{A}||_F^2 = \sum_{i,j} A_{ij}^2$.

matrix of $\mathbf{H}$, $\mathbf{C}$ does not change under the mean zero case of $\mathbf{H}$. Consequently, we could proceed our analyses under the dimensional normalized assumption without loss of generality. We turn to consider the dimensional normalized version of $V_\phi$ by replacing $\phi$ with $d_\phi$, which is $\phi$ with normalization:

$$d_\phi : \mathbb{R}^d \to \mathbb{R}^d, d_\phi(\mathbf{H}) = \phi(\frac{\sqrt{d}G\mathbf{S}^l\mathbf{H}}{||G\mathbf{H}||}), \tag{15}$$

where $G = \mathbf{I} - \frac{1}{d}\vec{1}\vec{1}^{T}$[4], i.e. $Gx = x - \mu\vec{1}, \quad \mu = \frac{1}{d}\sum_i x_i$.

**Theorem 2.** *Near the fixed point $\mathbf{C}^*$ of $V_{d_\phi}$, the exponential growth rate of the deviation of $\mathbf{C}^l$ from $\mathbf{C}^*$ is proportional to $K$.*

Here $\mathbf{C}^*$ is used to denote the BSB1 fixed points of $V_{d_\phi}(\mathbf{I}, \mathbf{C})$ [5]. Since $\mathbf{S}^l\mathbf{H}$ has covariance matrix $\mathbf{S}^l\mathbf{C}\mathbf{S}^l$, our scaling effect is that $V_{d_\phi}(\mathbf{S}, \mathbf{C}) = V_{B_\phi}(\mathbf{I}, \mathbf{S}^l\mathbf{C}\mathbf{S}^l)$. We use the following definitions to simplify notations.

**Define:** $$\mathbf{C}_G \triangleq GCG^T, K(\mathbf{S}, \mathbf{C}) \triangleq \mathbf{S}\mathbf{C}\mathbf{S}. \tag{16}$$

Theorem 3.6 of (Yang et al., 2019) tells us the derivative of $V_{d_\phi}(\mathbf{I}, \mathbf{C})$ (as a linear map) has a very explicit eigenspace decomposition, we describe it in Theorem 3. A simple reflection suggests that our linear operators still satisfy the DOS condition and the ultrasymmetry condition needed in the proof of this theorem, so this decomposition still holds.

**Theorem 3.** $J_{d_\phi} := \frac{dV_{d_\phi}}{d\mathbf{C}}$ *at $\mathbf{C}^*$ has eigenspaces and eigenvalues:*
*1. $V_0 = \{\mathbf{C}_0 : \mathbf{C}_G = 0\}$, with eigenvalue 0.* [6].
*2. $V_G = \mathbb{R}G$, with eigenvalue $\lambda_G$.*
*3. A $(d-1)$ dimensional eigenspace $V_L. = \{D^G : D \text{ diagonal}, trD = 0\}$.*
*4. A $\frac{d(d-3)}{2}$-dimensional eigenspace $V_M = \{\mathbf{C} : \mathbf{C}_G = \mathbf{C}, \text{ diag } \mathbf{C} = 0\}$ with eigenvalue $\lambda_M$.*
*$\lambda_L, \lambda_M < 1$, whereas $\lambda_G > 1$.*

So an appropriately chosen $\mathbf{S}$ can reduce the proportion that lies in $V_G$. We prove that the Frobenius norm of the component in $V_G$ is proportional to $K$ in Appendix H. Thus, it is natural to consider the orthogonal (in terms of Frobenius norm and corresponding inner product) eigendecomposition (with subindices indicating the corresponding eigenspaces we listed above):

$$K(\mathbf{S}, \mathbf{C}) = \mathbf{C}_0 + \mathbf{C}_G + \mathbf{C}_L + \mathbf{C}_M. \tag{17}$$

The effect of Dr is to reduce the $\mathbb{R}G-$component at each step to make the dynamic system more stable. Since the decomposition is orthogonal, this is equivalent to reducing

$$G_l :=< \mathbf{S}^l\mathbf{C}\mathbf{S}^l, G >, \tag{18}$$

recall that $G = \mathbf{I} - \frac{1}{d}\vec{1}\vec{1}^{T}$, i.e. $Gx = x - \mu\vec{1}, \quad \mu = \frac{1}{d}\sum_i x_i$. Since we take the normalization assumption, only the relative magnitude of $s_{li}$ matters, and we can put any homogeneous restriction. In order to include the case $s_{li} = 1$, we consider the restriction $\sum_{i=1}^d s_{li}^2 = d$. By definition $\mathbf{C}_{ij} = E[h_i h_j]$, hence we have

$$< \mathbf{S}^l\mathbf{C}\mathbf{S}^l, G > = Tr(\mathbf{S}^l\mathbf{C}\mathbf{S}^l(\mathbf{I} - \frac{1}{d}(\vec{1}\vec{1}^T)^T)$$
$$= Tr(\mathbf{S}^l\mathbf{C}\mathbf{S}^l) - \frac{1}{d}Tr(\mathbf{S}^l\mathbf{C}\mathbf{S}^l\vec{1}\vec{1}^T) \tag{19}$$
$$= \sum_i c_{ii}s_{li}^2 - \frac{1}{d}\sum_{i,j} c_{ij}s_{li}s_{lj}.$$

Finally we come to the effectiveness measure of the proposed Dr Block.

$$K = \frac{\sum_i c_{ii}s_{li}^2 - \frac{1}{d}\sum_{i,j} c_{ij}s_{li}s_{lj}}{(\sum_i c_{ii} - \frac{1}{d}\sum_{i,j} c_{ij}) \times \frac{\sum_i s_{li}^2}{d}}. \tag{20}$$

---

[4]$\vec{1}$ is the $d-$dimensional vector with all component 1.

[5]All results involve the BSB1 fixed point (Yang et al., 2019) require permutation, diagonal and off-diagonal symmetry, and hold for dimensional normalization, too.

[6]$G$, $\mathbf{S}$ above are symmetric, so the transpose is only introduced for the sake of notational balance

Table 1: Dataset statistics.

|  | CoraR | CiteseerR | Pubmed | PPI | Reddit | A* |
|---|---|---|---|---|---|---|
| Nodes/Users | 2,680 | 3,191 | 19,717 | 56,944 | 232,965 | 35,246,808 |
| Edges | 5,148 | 4,172 | 44,324 | 818,716 | 11,606,919 | 129,834,116 |
| Classes/Items | 7 | 6 | 3 | 121 | 41 | 6,338,428 |
| Features | 302 | 768 | 500 | 50 | 602 | 27 |
| Training Nodes | 1,180 | 1,691 | 18,217 | 44,906 | 152,410 | 35,246,808 |
| Validation Nodes | 500 | 500 | 500 | 6,514 | 23,699 | - |
| Test Nodes | 1,000 | 1,000 | 1,000 | 5,524 | 55,334 | 35,246,808 |

The denominator is chosen to display the ratio of variance reduction for the proposed Dr Block. Without Dr, $s_{li} = 1$ for all $i$, and we have $K = 1$. From our calculation on the inner product, it can be discovered that this quantity is proportional to the part in $V_G$ in the orthogonal decomposition, this proves Theorem 2. Since this is the only part for $J_{d_\phi}$ with eigenvalue larger than 1, the exponential growth rate is proportional to this quantity. Therefore, this quantity measures the magnitude of improvement Dr Blocks make to the stability of the learning process under perturbation.

## 4 EXPERIMENTS

In this section, we evaluate the proposed DrGCNs on a variety of datasets compared to several SOTA methods. Detailed descriptions of the experiments and datasets are included in Appendix C,D.

### 4.1 EXPERIMENTAL SETTINGS

**Datasets** We present the performance of DrGCNs on several public benchmark node classification datasets, including Pubmed (Yang et al., 2016), Reddit, PPI (Hamilton et al., 2017). We also conduct experiments on a large-scale real-world commercial recommendation A* dataset. Table 1 summarizes statistics of the datasets.

There are also two widely adopted Cora and Citeseer datasets (Yang et al., 2016) for citation networks. We investigate the originality of these datasets not only from the public data provided by (Yang et al., 2016) but also from much earlier versions (McCallum et al., 2000; Lu & Getoor, 2003) and find problems in those datasets. 32(1.2%) samples in Cora and 161(4.8%) samples in Citeseer are duplicated, while 1,145(42.3%) samples in Cora and 2,055(61.8%) samples in Citeseer have information leak that includes their labels directly as a dimension of their features. To address such problems, we remove duplicated samples, modify their features using word and text embeddings to reduce information leak, and construct two refined datasets, CoraR and CiteseerR. For details of these refined datasets and A* dataset, please refer to Appendix A,B,E.

**Competitors** We compare the results of our DrGCN with GCN (Kipf & Welling, 2017), GAT (Veličković et al., 2018), MixHop (Abu-El-Haija et al., 2019), GraphSAGE (Hamilton et al., 2017), FastGCN (Chen et al., 2018) and ASGCN (Huang et al., 2018) on citation networks including CoraR, CiteseerR and Pubmed. We also provide results on the original Cora and Citeseer dataset in Appendix C. Our DrGCN also works for inductive datasets that we evaluate the Dr-GAT with several state-of-the-art methods on the PPI dataset, including GraphSAGE (Hamilton et al., 2017), LGCN (Gao et al., 2018), and GeniePath (Liu et al., 2019). As for A* dataset, we compare our method with the company's previous best GraphSAGE model, see Appendix E.

**DrGCNs** We combine Dr block with five most representative GCN methods and compare with them on public datasets. Two full GCN methods include the vanilla GCN (Kipf & Welling, 2017) and a variant that exploits an attention aggregator GAT (Veličković et al., 2018). Sampling GCN methods contain FastGCN (Chen et al., 2018), Adaptive Sampling GCN (Huang et al., 2018), and A* company's heterogeneous GraphSAGE model on A* dataset. Every GCN layer is replaced by a DrGCN layer as in Equation (4). Further implementation details are covered in Appendix C.

### 4.2 RESULTS AND ANALYSES

Table 2 illustrates the performance of DrGCNs on four transductive datasets when combined with four different variations of GCN models. Our results are averaged among 20 runs with different random seeds. Our DrGCNs achieve superior performances on all of the datasets and demonstrate

Table 2: Summary of classification accuracy on public transductive datasets(%).

| Category | Method | CoraR | CiteseerR | Pubmed | Reddit |
|---|---|---|---|---|---|
| Full GCNs | GCN | 85.9±0.5 | 74.9± 0.7 | 88.0± 0.3 | - |
| | GAT | 86.9± 0.4 | 76.5± 0.4 | 85.0± 0.2 | - |
| | Mix-Hop | 85.9± 0.5 | 75.3± 0.3 | 88.1± 0.2 | - |
| Sampling-based GCNs | GraphSage | 84.9± 0.5 | 70.6± 0.8 | 84.2± 0.3 | 94.8± 0.1 |
| | FastGCN | 81.7± 0.4 | 74.7± 0.6 | 88.3± 0.4 | 92.5± 0.2 |
| | ASGCN | 84.3± 0.7 | 75.1± 0.9 | 89.8± 0.3 | 96.4± 0.3 |
| DrGCNs | Dr-GCN | 86.8± 0.5 | **77.5± 0.6** | 88.4± 0.3 | - |
| | Dr-GAT | **87.3± 0.2** | 77.0± 0.3 | 84.7± 0.4 | - |
| DrGCNs(sampling based) | Dr-FastGCN | 81.6± 0.5 | 75.5± 0.5 | 88.2± 0.3 | 94.0± 0.1 |
| | Dr-ASGCN | 84.3± 0.5 | 75.4± 0.4 | **90.3± 0.4** | **97.9± 0.1** |

Table 3: Summary of performance (micro F1) on inductive PPI dataset.

| Method | GraphSAGE | LGCN | GeniePath | GAT | Dr-GAT |
|---|---|---|---|---|---|
| micro F1 | 61.2 | 77.2 | 97.9 | 97.3 | **98.8± 0.1** |

relatively significant improvement on the Reddit dataset. Dr-ASGCN even reduces the error rates by more than $40\%$ ($3.6\% \rightarrow 2.1\%$), compared with previous state-of-the-art method ASGCN.

The performance improvements can be explained by our stability measure proposed in Equation (20). Theoretically, when $K \approx 1$, we expect Dr block to have limited ability in refining the representation, while when $K \ll 1$ we expect the vector to strengthen the stability of the model by reducing the magnitude of the derivatives of the covariance matrix and improve the performance. To verify the theoretical analyses, we collect the average $K$-value of the learnt reweighting vectors for different layers in the Dr-ASGCN model, see Table 5. The K-value in the second layer is around 1 on all datasets. However, the K-value for the first layer is around 1 for citation datasets, but 0.32 on the Reddit dataset, which emphatically explains why the DR-ASGCN achieves such a massive improvement on the Reddit dataset.

On the inductive PPI dataset (Table 3), Dr Block increases the micro f1-score of GAT by $1.5\%$ and outperforms all previous methods. Table 4 suggests that the Dr method can also accomplish substantial improvements on the real-world, large-scale recommendation dataset. It demonstrates improvement on industrial measure recall@50, which is the rate of users clicking the top 50 predicted items among 6 million different items within the next day of the training set, from $5.19\%$ (previous best model) to $5.26\%$ (Dr Block added).

### 4.3 BATCH-NORM AND LAYER-NORM

We also compare DrGCNs with other feature refining methods, including Batch-Norm (Ioffe & Szegedy, 2015) and Layer-Norm (Lei Ba et al., 2016). These methods use variance information on every single dimension to refine representations, while DrGCN joins information on each dimension and learns a reweighting vector **S** adaptively. We provide results of DrGCN and these methods on the Reddit dataset for ASGCN( Table 6). Batch-Norm and Layer-Norm also improve the performance of the ASGCN model on the Reddit dataset. Combining Dr and Layer-norm yields an even better result for ASGCN on Reddit. More detailed results are in Appendix F.

### 5 RELATED WORKS

The idea of using neural networks to model graph-based data can be traced back to Graph Neural Network (Scarselli et al., 2009), which adopts a neural network structure on graph structure learning. GCN (Kipf & Welling, 2017) proposes a deep-learning-based method to learn node representations on a graph using gathered information from the neighborhood of a node. GraphSAGE (Hamilton et al., 2017) formulated a sample and aggregation framework of inductive node embedding. The idea of the sample and aggregation framework is to incorporate information from the neighborhood to generate node embeddings. Despite being uniform when first being proposed, both sampling and aggregation can be weighted. These methods, including FastGCN (Chen et al., 2018), GAT (Veličković

Table 4: Performance (Recall@50) on A* online recommendation system. GraphSAGE refers to A* company's best heterogeneous GraphSAGE model.

| Method | GraphSAGE | DrGraphSAGE |
|---|---|---|
| Recall@50(%) | 5.19 | **5.26** |

Table 5: Classification accuracy of ASGCN and Dr-ASGCN model, with average K value learnt for each layer.

| Method | Cora | Citeseer | CoraE | CiteseerE | Pubmed | Reddit |
|---|---|---|---|---|---|---|
| ASGCN | 87.23 | 78.95 | 84.30 | 75.12 | 89.82 | 96.37 |
| Dr-ASGCN | 87.07 | 79.06 | 84.34 | 75.44 | 90.34 | 97.95 |
| Improvement Rate(%) | -0.16 | 0.11 | 0.04 | 0.32 | 0.52 | 1.58 |
| Learnt K-value(Layer 1) | 1.04 | 1.01 | 0.90 | 0.95 | 0.98 | 0.32 |
| Learnt K-value(Layer 2) | 1.00 | 1.00 | 0.99 | 1.04 | 0.98 | 1.14 |

Table 6: Accuracy and average training time per epoch for plain, batchnorm, layernorm and DrAS-GCN methods on Reddit dataset.

| Method | ASGCN | Batch-norm | Layer-norm | Dr | Dr+LN |
|---|---|---|---|---|---|
| Accuracy(%) | 96.37$\pm$ 0.22 | 96.99$\pm$ 0.15 | 97.68$\pm$ 0.15 | 97.95$\pm$ 0.13 | **98.02$\pm$ 0.12** |
| Time(s/epoch) | 17.99 | 17.90 | 18.74 | 17.46 | 17.93 |

et al., 2018), LGCN (Gao et al., 2018), ASGCN (Huang et al., 2018), GaAN (Zhang et al., 2018), and Mix-Hop (Abu-El-Haija et al., 2019), treat all nodes in the graph unequally and try to figure out more important nodes and assign them higher weights in sampling and aggregation procedure.

Feature imbalance phenomena have long been aware of. (Blum & Langley, 1997) Different dimensions of the hidden representation generated by neural networks may also share such imbalance behavior. The idea of refining hidden representations in neural networks can be traced back to Network in Network (Lin et al., 2014), whom proposes a fully-connected neural network to refine the pixel-wise hidden representation before each convolutional layer–known as the $1 \times 1$ convolution layer which is widely adopted in modern convolutional neural networks. Squeeze and Excitation Networks (Hu et al., 2018) proposes a dimensional reweighting method called Squeeze and Excitation block, which involves the techniques of global average pooling and encoder-decoder structure. It works well in computer vision CNNs and wins the image classification task of Imagenet 2017. The success attracts our concern that such dimensional reweighting methods might also be useful in node representation learning on graphs.

Another natural idea to refine representations of neural networks is normalization. Batch normalization (Ioffe & Szegedy, 2015) is a useful technique in neural networks to normalize and reduce the variance of input representations. Layer normalization (Lei Ba et al., 2016) is an improved version of BatchNorm, for it also works on a single sample.

Many also try to give theoretical analyses to such normalization techniques. (Kohler et al., 2018) explains the efficiency of batch normalization in terms of convergence rate. (Bjorck et al., 2018) shows that batch normalization enables larger learning rates. (Yang et al., 2019) demonstrates the gradient explosion behaviors of batch normalization on fully-connected networks via mean field theory (Kadanoff, 2009). In our approach, we adopt some of these methods and apply them to the analysis of DrGCNs.

# 6 CONCLUSION

We propose **D**imensional **r**eweighting **G**raph **C**onvolutional **N**etworks (DrGCNs) and prove that DrGCNs can improve the stability of GCN models via mean field theory. Further explorations lead to the proposal of a new measure K to evaluate the effectiveness of DrGCNs. We conduct experiments on various benchmark datasets and compare DrGCNs with several GCN variations. Experimental results not only prove the efficiency of DrGCNs, but also support the theoretical analysis on measure K. DrGCNs' usefulness is likewise verified on large-scale industrial dataset A*.

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

# Appendices

## A  PROBLEMS OF CORA AND CITESEER DATASET

We investigate the originality of the Cora and CiteSeer dataset. The two datasets are widely used for being light-weighted and easy to handle. The most popular version is provided by Planetoid (Yang et al., 2016). The two datasets are both citation networks where each node represents a research paper, and each edge represents a citation relationship between two papers. Edges are directed but are usually handled undirectedly by GCN methods. Each paper belongs to a sub-field in computer science and is marked as its label. Papers have features of bag-of-word(BOW) vectors that each dimension represents whether the document of the paper contains a particular word in the dictionary or not. Cora has 2,708 papers with a dictionary size of 1,433, while Citeseer has 3,327 papers with a dictionary size of 3,703.

### A.1  CORA

Cora originates in (McCallum et al., 2000)[7] with extracted information(including titles, authors, abstracts, references, download links etc.) in plain-text form. Those download links are mostly unavailable now. Before they become unavailable, (Lu & Getoor, 2003)[8] extracts a subset of 2,708 papers and assigns labels and BOW feature vectors to the papers. The dictionary is chosen from words(after stemming)[9] that occur 10 or more times in all papers and result in a dictionary size of 1,433. Planetoid (Yang et al., 2016) reordered each node to form the benchmark Cora dataset (Yang et al., 2016).

There exist a lot of duplicated papers (one paper appears as multiple identical papers in the dataset) in the original Cora of (McCallum et al., 2000), and (Lu & Getoor, 2003) inherits the problem of duplicated papers. In Cora, we find 32 duplicated papers among the 2,708 papers. Another problem is the information leak. The generation process of the dictionary chooses words that occur more than 10 times, and does not exclude the label contexts of papers. Therefore, some papers may be classified easily only by looking at their labels. For instance, $61.8\%$ of papers labeled "reinforcement learning" contain exactly the word "reinforcement" "learning" in their title and abstract(after stemming). Altogether 1,145($42.3\%$) of these papers contain their label as one or some of the dimensions of their features. [10]

### A.2  CITESEER

CiteSeer is a digital library maintained by PSU(Currently named as CiteSeerX[11]), which provides a URL based web API for each paper according to their doc-ids. (Lu & Getoor, 2003)[12] extracts 3,327 nodes from the library to form a connected citation network in which 3,312 nodes are associated with a BOW vector of dictionary size 3,703. The rest 15 nodes' features are padded with zero vectors. It is also reordered and adopted by Planetoid (Yang et al., 2016).

Although the original version of Citeseer only consists of links that are unavailable now, a back-up version of Citeseer contains the title and abstract information of 3,186 of the papers. [13] We also find another 81 by author and year information using Google Scholar. Unfortunately, among the papers we collected from the Citeseer dataset, 161($4.8\%$) of them are actually duplicated.

Since the Citeseer dataset shares a similar embedding generation method with Cora, there also exists the problem of information leak. For the data we collected, at least $2,055(61.8\%)$ of the papers in Citeseer contain their labels in title or abstract, which are sure to become some of the dimensions of their feature representations.

---

[7]https://people.cs.umass.edu/~mccallum/data/cora.tar.gz

[8]https://linqs-data.soe.ucsc.edu/public/lbc/cora.tgz

[9]Stemming basically transforms a word into its original form.

[10]If we include references of the paper the rate becomes $86.1\%$, see what GCN learns!

[11]http://citeseerx.ist.psu.edu

[12]https://linqs-data.soe.ucsc.edu/public/lbc/citeseer.tgz

[13]https://www.cs.uic.edu/~cornelia/russir14/lectures/citeseer.txt

Table 7: Label terms appearing in Cora's title and abstract.

|  | Case Based | Theory | Probabilistic Methods | Rule Learning | Neural Networks | Reinforcement Learning | Genetic Algorithms | Total |
|---|---|---|---|---|---|---|---|---|
| Total | 298 | 351 | 426 | 180 | 818 | 217 | 418 | 2708 |
| Appeared | 151 | 84 | 103 | 38 | 410 | 134 | 225 | 1145 |
| Rate | 50.7% | 23.9% | 24.2% | 21.1% | 50.1% | 61.8% | 53.8% | 42.3% |

Table 8: Label terms appearing in CiteSeer's title and abstract.

|  | Artificial Intelligence | Machine Learning | Information Retrieval | Database | Agents | Human Computer Interface | Total |
|---|---|---|---|---|---|---|---|
| Total | 264 | 590 | 668 | 701 | 596 | 508 | 3327 |
| Appeared | 54 | 334 | 434 | 403 | 521 | 309 | 2055 |
| Rate | 20.5% | 56.6% | 65.0% | 57.5% | 87.4% | 60.8% | 61.8% |

# B    THE REFINED CORAR AND CITESEERR DATASET

Due to the problems in Cora and Citeseer, we produce new datasets CoraR and CiteSeerR to address these issues. We remove duplicated nodes and generate indirect pre-trained word/text embeddings as node features to reduce the information leak.

## B.1    CORAR

We double-check the articles belong to Cora (Lu & Getoor, 2003) dataset in the original Cora (Mc-Callum et al., 2000). Among the 2,708 papers, 474 of them have a wrong title that can hardly be found on any academic search engines such as google scholar or aminer (Tang et al., 2008). We manually searched for these papers based on other information like author, abstract, references and feature vectors in Cora. Finally, we figure out that 32 of the total 2,708 papers are duplicated papers that actually belong to 13 identical ones. 9 papers are absolutely missing and not able to trace. We recover the actual title of the rest 2,680 papers, and use their titles to generate their features. We apply averaged GLOVE-300 (Pennington et al., 2014) word vector for their titles(with stop words removed) and we add two dimensions expressing the length of the title, and the length of the title with stop words removed. This leads to a 302 dimension feature representation for each node in CoraR. The average and word embedding process can better reduce the effect of label information leak than using simple BOW vectors. As in (Hamilton et al., 2017; Chen et al., 2018), we split the 2,680 nodes into a training set of 1,180 nodes, a validation set of 500 nodes and a test set of 1,000 nodes.

## B.2    CITESEERR

We use Cornelia's back-up extractions of CiteSeer and manually find some other documents using author and year information from academic search engines. For the rest we only have a numerical ID and an invalid download link, so we are not able to trace them. We check the titles, abstracts and feature vectors and find 161 papers are actually 80 identical papers. We combine duplicated papers together and also remove 55 papers that we are not able to trace. Our refined CiteSeerR has 3,191 nodes and 4,172 edges. We average over the last hidden state of BERT (Devlin et al., 2019) for each sentence from the title and abstract and produce a 768-dimensional feature for each paper. For pairs of duplicated papers with distinct labels, we manually assign a label for it. As in (Hamilton et al., 2017; Chen et al., 2018), we split the 3,191 nodes into a training set of 1,691 nodes, a validation set of 500 nodes and a test set of 1,000 nodes.

Both CoraR and CiteseerR are included in the link of code.

# C    IMPLEMENTATION DETAILS

## C.1    DR BLOCK CONFIGURATION

For the dimensional reweighting part of each layer, the dimension of the encoder $\mathbf{g}$ is set to the closest integer of the square root of the dimension of the input node representation $\mathbf{r}^i$. The pooling weight $w_v$ is set to uniform weight $w_v = \frac{1}{|\mathcal{V}|}$, where $|\mathcal{V}|$ is the number of nodes in the graph for full GCNs, and the batch size for batch-wise sampling GCNs. We use ELU activation for $\sigma_g$ and sigmoid activation for $\sigma_s$. We do not apply dropout, but apply weight regularization with the same coefficient as the original GCNs in the Dr Blocks of DrGCNs.

We keep all other settings, including learning rate, early stop criteria, loss function, hidden representation dimension, batch size, weight decay, and regularization loss the same as the original models.

## C.2    EVALUATION DETAILS

All of our methods and compared baseline methods are run 20 times, and we report the average accuracy and variation for methods that we run. We evaluate their performance mainly based on their released code and paper. For methods without codes, we use their reported performance if they share the same evaluation settings as ours.

## C.3    METHOD DETAILS

We describe our method implementation details here.

**GCN** (Kipf & Welling, 2017) We use the GCN code provided in AS-GCN[14], and use a 2-layer GCN model with 64 hidden units in layer 1. We run it 20 times and report the average and variance. For each running time, we use the model that performs best within 800 training epochs on the validation set for testing.

**GAT** (Veličković et al., 2018)

We use the GAT code provided by the authors [15]. We use the dataset-specific structures described in (Veličković et al., 2018) and early stopping strategy mentioned in the code repo. The original paper uses a high dropout rate of 0.6 on semi-supervised citation datasets test setting. We find that for CoraR, CiteseerR and the fully-supervised training set setting on Pubmed, such a high dropout may have a chance to lead the original GAT model not to converge, so we adjusted the dropout rate to 0.35(which gives the best performance among all dropout rates from 0 to 0.6) for both the original and our Dr-GAT. On PPI we simply follow their instructions and use their suggested structure and hyperparameters.

GAT forms a 2 layer structure for citation datasets. For Cora and Citeseer, GAT has 8 hidden units in every attention head in the first layer, and 8 attention heads in the first layer and 1 in the second layer, which has the number of hidden units equal to node classes. We adopt the structure for CoraR and CiteseerR. We also discover that for the fully-supervised training set setting on Pubmed, the structure for Pubmed in GAT paper(which has 8 heads in the second layer) does not perform as good as the GAT structure for Cora and Citeseer (this setting achieves $82.5 \pm 0.3\%$ under the best dropout rate), so we also adopt the Cora/Citeseer structure to Pubmed.

For PPI, GAT has a three layer structure, with 256 hidden units in every attention head in the first two layers. It has 4 attention heads in the first layer and 4 in the second layer, and 6 attention heads in the third layer, each third layer attention head has a number of hidden units equal to node classes. It also sets dropout equals to 0 and uses residual connection (He et al., 2016).

---

[14]`https://github.com/huangwb/AS-GCN`
[15]`https://github.com/PetarV-/GAT`

**FastGCN** (Chen et al., 2018) We run the code provided by the authors. [16] We use the weighted sampling method described in the paper, and use a neighborhood size of 400 for CoraR, CiteseerR, Pubmed and 512 for Reddit. We run 20 times and generate an average performance and variations.

**ASGCN** (Huang et al., 2018) We use the code provided by the authors [17]. We use a neighborhood size of 128 for CoraR, CiteseerR, 256 for Pubmed and 512 for Reddit as the paper suggested. (Huang et al., 2018) Their original code seems to have a bug causing unnecessary test set information leaking during the validation process. We modified their code to avoid such problems. We choose the best model on validation set within 800 epochs for citation datasets and 200 epochs for Reddit and use that for testing.

**Mix-Hop** (Abu-El-Haija et al., 2019) We use the code provided by the authors [18]. We use the best author provided script for Pubmed. Unfortunately the parameters the authors provided for Cora and Citeseer does not work well for CoraR and CiteseerR, so we fix the network structure and tune the hyperparameter set for CoraR and CiteseerR ourselves.

**GraphSAGE** (Hamilton et al., 2017) We use the code provided by the authors[19] on CoraR, CiteseerR and Pubmed. For Reddit this code does not work so we use another code also provided by the author [20]

**Other Methods** Many of them do not have their codes released, so we use their reported performance in their papers, or reported performance of their method by other papers, if we share a similar evaluation setting.

Specifically, the performances of all baseline methods in appendix table 11 are from their original papers (some non-GCN baseline results are from (Kipf & Welling, 2017)). The performance of GraphSAGE, GeniePath, LGCN, GAT on the PPI dataset are from their original papers.

# D   DATASETS

The details of our datasets are listed in this section. We generally use 6 datasets, including 3 citation datasets, 1 Reddit dataset, 1 inductive PPI dataset, and 1 sizeable online recommendation A* dataset.

**Citation Networks** We evaluate the performance of our DR models on the three citation network datasets, CoraR, CiteseerR, and Pubmed (Sen et al., 2008; Yang et al., 2016) There are two types of experimental settings, the semi-supervised setting for full GCNs (Kipf & Welling, 2017; Veličković et al., 2018), which uses only a little fraction of the node labels on the graph and all link information for training. Another fully-supervised setting (Chen et al., 2018; Huang et al., 2018) uses node labels of the full graph except the validation and test set for training. The dataset statistics of original Cora, Citeseer, and Pubmed dataset under these two settings are shown in table 9. We double-check the adjacency matrices, remove self-loops, and correct the number of edges in these datasets, which is mistaken in various papers, including GCN (Kipf & Welling, 2017). Besides evaluation on CoraR, CiteseerR, and Pubmed in the main article, we provide the result of the fully-supervised setting for Cora and Citeseer in table 10. We also provide our results on the semi-supervised setting on Cora, Citeseer, and Pubmed dataset in table 11.

**PPI** The protein-protein interaction dataset is collected by SNAP (Hamilton et al., 2017) from the Molecular Signatures Database (Subramanian et al., 2005), which is an inductive multi-label node classification task. The training set contains 20 protein graphs, while the validation and test set contains two graphs each. We evaluate the performance of different models by micro F1-score.

**Reddit** The Reddit dataset is collected by SNAP (Hamilton et al., 2017) from Reddit posts. It is a node classification dataset for classifying different communities of each user by his/her posts.

---

[16]https://github.com/matenure/FastGCN
[17]https://github.com/huangwb/AS-GCN
[18]https://github.com/samihaija/mixhop
[19]https://github.com/williamleif/graphsage-simple
[20]https://github.com/williamleif/GraphSAGE

Table 9: Supplement of Dataset statistics for citation datasets.

|  | Cora | Citeseer | Pubmed |
|---|---|---|---|
| Nodes | 2,708 | 3,327 | 19,717 |
| Edges | 5,278 | 4,552 | 44,324 |
| Classes | 7 | 6 | 3 |
| Features | 1,433 | 3,703 | 500 |
| Training Nodes(Semi) | 140 | 120 | 60 |
| Training Nodes(Full) | 1,208 | 1,827 | 18,217 |
| Validation Nodes | 500 | 500 | 500 |
| Test Nodes | 1,000 | 1,000 | 1,000 |

Table 10: Summary of classification accuracy on fully supervised Cora and Citeseer(%).

| Category | Method | Cora | Citeseer |
|---|---|---|---|
| Full GCNs | GCN Kipf & Welling (2017) | 86.4±0.3 | 77.4± 0.2 |
|  | GAT Veličković et al. (2018) | 87.2± 0.4 | 77.8± 0.2 |
| Sampling-based GCNs | FastGCN Chen et al. (2018) | 83.9± 0.4 | 78.6± 0.4 |
|  | ASGCN Huang et al. (2018) | 87.2± 0.2 | 79.0± 0.4 |
| Dr GCNs(ours) | Dr-GCN | 86.8± 0.2 | 77.5± 0.3 |
|  | Dr-GAT | **87.4±0.2** | 77.8± 0.2 |
| Dr Sampling-based GCNs(ours) | Dr-FastGCN | 84.0± 0.4 | 78.3± 0.3 |
|  | Dr-ASGCN | 87.1± 0.2 | **79.1± 0.4** |

## E   A* DATASET: DATASET, BASELINE AND EVALUATION

As for the industrial A* dataset we use. It is an item recommendation dataset, with the training set has about 35 million users and 6.3 million items with 120 million edges. Although the target is node-classification like(to find the most likely items that each user may click), instead of simply taking each item as a class, A* uses a graph embedding model to generate embedding for both users and items. There are 27 user attributes and 33 item attributes. For every user, we use K nearest neighbor (KNN) with Euclidean distance to calculate the top-N items that the user is most likely to click, and the customer will see these recommended items in A* company's APP. We use the recall@N to evaluate the model:

$$recall@N = mean(\sum_u \frac{|M_u| \cap |I_u|}{|I_u|})$$ (21)

$M_u$ represents the top-N items recommended by the model and $I_u$ indicates the items clicked by the customer. The baseline model is the A* online heterogeneous GraphSAGE, and we add Dr block in it to compare Recall@N with the online model.

Recall@50 is the most commonly used metric in A* company. Experimental results show that we reach $5.264\%$ on Recall@50, improving from the original best model's $5.188\%$. It is quite a good result, considering random guess will only give less than $0.001\%$ (50/6,300,000).

## F   BATCH-NORM AND LAYER-NORM GCNS ON CITATION NETWORKS

In Table 12 we also provide the batch-norm and layer-norm GCN results on publication datasets. The results are averaged among 20 runs.

## G   PROOF OF THEOREM 1

We also provide proof for theorem 1 in the main article.

Table 11: Summary of classification accuracy (%) of semi-supervised labels on citation datasets.

| Category | Method | Cora | Citeseer | Pubmed |
|---|---|---|---|---|
| Non-Graph Convolution | MLP | 55.1 | 46.5 | 71.4 |
| | ManiReg (Belkin et al., 2006) | 59.5 | 60.1 | 70.7 |
| | SemiEmb (Weston et al., 2012) | 59.0 | 59.6 | 71.1 |
| | LP (Zhu et al., 2003) | 68.0 | 45.3 | 63.0 |
| | DeepWalk (Perozzi et al., 2014) | 67.2 | 43.2 | 65.3 |
| | ICA (Lu & Getoor, 2003) | 75.1 | 69.1 | 73.9 |
| | Planetoid (Yang et al., 2016) | 75.7 | 64.7 | 77.2 |
| | GraphSGAN (Ding et al., 2018) | 83.0 | **73.1** | – |
| Graph Convolution | Chebyshev (Defferrard et al., 2016) | 81.2 | 69.8 | 74.4 |
| | GCN (Kipf & Welling, 2017) | 81.5 | 70.3 | 79.0 |
| | MoNet (Monti et al., 2017) | 81.7 | – | 78.8 |
| | DPFCNN (Monti et al., 2018) | 83.3 | 72.6 | – |
| | LGCN (Gao et al., 2018) | 83.3 | 73.0 | 79.5 |
| | GAT (Veličković et al., 2018) | 83.0 | 72.5 | 79.0 |
| | Mix-Hop (Abu-El-Haija et al., 2019) | 81.9 | 71.4 | **80.8** |
| **DR**-GCNs(ours) | DR-GCN | 81.6± 0.1 | 71.0±0.6 | 79.2±0.4 |
| | DR-GAT | **83.6± 0.5** | 72.8± 0.8 | 79.1±0.3 |

Table 12: GCN, BatchNorm , LayerNorm and DrGCN methods accuracy(%) on citation datasets.

| Method | CoraR | CiteseerR | Pubmed |
|---|---|---|---|
| GCN | 85.9± 0.5 | 74.9± 0.7 | 88.0± 0.3 |
| Batch-norm | 86.2± 0.5 | 77.5±0.4 | 88.6 ± 0.3 |
| Layer-norm | 85.6± 0.7 | 76.4 ± 0.9 | 89.9± 0.4 |
| Dr-GCN | 86.8± 0.5 | 77.5± 0.6 | 88.4± 0.3 |

Now we turn to prove theorem 1. We say a linear operator $T : H_d \to H_d$ is diagonal-off-diagonal semidirect (DOS) if and only if:

$$\forall \mathbf{C} \in H_d, T(\mathbf{C})_{ii} = uc_{ii},$$
$$T(\mathbf{C})_{ij} = vc_{ii} + vc_{jj} + wc_{ij}.$$

Here $u, v, w$ are constants, and we will call the set of operators with these parameters $DOS(u, v, w)$. By the definition of $V_\phi$, the $(i, j)$ component of its output will only involve the $i - th$ and $j - th$ components of the input and symmetric with respect to them, hence itself and its derivatives $J_\phi$ will also involve them only and being symmetric with respect to them. Thus it is determined by $c_{ii}, c_{ij}, c_{jj}$. Furthermore, since $J_\phi$ is a linear map, so it will have this form. The result in Theorem 1 should hold in general for $DOS$ operators and do not require information about the fixed point structure.

Now $J_\phi$ is a DOS operator, hence it will belong to $DOS(u, v, w)$ for some $u, v, w$. Then we know:

$$||J_\phi(\mathbf{C})||_F^2 = \sum_{i,j=1}^{d} (J_\phi(\mathbf{C})_{ij})^2$$
$$= \sum_{i=1}^{d}(uc_{ii})^2 + \sum_{i \neq j}(vc_{ii} + vc_{jj} + wc_{ij})^2.$$

Correspondingly we have:

$$||J_\phi(\mathbf{SCS})||_F^2 = \sum_{i,j=1}^d (J_\phi(\mathbf{SCS})_{ij})^2$$

$$= \sum_{i=1}^d (us_i^2 c_{ii})^2 + \sum_{i \neq j} (vs_i^2 c_{ii} + vs_j^2 c_{jj} + ws_i s_j c_{ij})^2.$$

Since the inequality we want to prove is homogeneous of degree 2 with respect to $s_i$ on both sides, hence without loss of generality we can assume $\sum_{i=1}^d s_i^2 = d$ (this choice of gauge is intended to include the case in which all $s_i = 1$). Consider the function of (nonzero) $\mathbf{C} \in H_d$:

$$K(\mathbf{C}) := min_{\mathbf{S}} \frac{||J_\phi(\mathbf{SCS})||_F^2}{||J_\phi(\mathbf{C})||_F^2}.$$

Here $min_{\mathbf{S}}$ is minimizing over diagonal $\mathbf{S}$ with $\sum_{i=1}^d s_i^2 = d$, which is a compact set, hence the minimum is achieved at some point for any fixed $\mathbf{C}$. And notice that at $\mathbf{S} = Id$, $K = 1$, so $K(\mathbf{C}) \leq 1$ and the equality will hold if and only if $\mathbf{S} = Id$ is the minimun point of:

$$\tilde{K}(\mathbf{C}, \mathbf{S}) : = ||J_\phi(\mathbf{SCS})||_F^2$$

$$= \sum_{i=1}^d (us_i^2 c_{ii})^2 + \sum_{i \neq j} (vs_i^2 c_{ii} + vs_j^2 c_{jj} + ws_i s_j c_{ij})^2,$$

with this fixed $\mathbf{C}$. In particular, we know that at $s_1 = s_2 = ... = s_d = 1$ this function satisfties Karush-Kuhn-Tucker (KKT) conditions (which could also be derived from the method of Lagrange multiplier in this case). Next we derive the KKT condition for each component. Define:

$$L(s_1, ..., s_d, \lambda) \triangleq \sum_{i=1}^d (us_i^2 c_{ii})^2 + \sum_{i \neq j} (vs_i^2 c_{ii} + vs_j^2 c_{jj} + ws_i s_j c_{ij})^2 - \lambda (\sum_{i=1}^d s_i^2 - d),$$

then the KKT conditions (i.e., the extreme value condition for restricted optimization) are:

$$4u^2 s_i^3 c_{ii} + \sum_{j \neq i} 2(vs_i^2 c_{ii} + vs_j^2 c_{jj} + ws_i s_j c_{ij})(2vs_i c_{ii} + ws_j c_{ij}) - 2\lambda s_i = 0.$$

Now evaluate at $s_1 = ... = s_d = 1$, we have:

$$4u^2 c_{ii} + \sum_{j \neq i} 2(vc_{ii} + vc_{jj} + wc_{ij})(2vc_{ii} + wc_{ij}) - 2\lambda = 0.$$

When $v \neq 0$, the coefficient of $\mathbf{C}^2$ is nonzero. Thus this gives a quadratic defining function for those $\mathbf{C}$ where our statement may fail. Denote the left hand side of the equation by $F_i$, when $\nabla_{\mathbf{C}} F_i \neq 0$, it defines a smooth codimension 1 submanifold of $H_d$. When $\nabla_{\mathbf{C}} F_i = 0$, it gives rise to a linear equation, in which $c_{ii}$ has coefficient $4(d-1)v^2$, hence still gives rise to a codimension one smooth submanifold of $H_d$. In particular, the union of them is a codimension 1 object (not necessarily smooth after we take union). Therefore, those $\mathbf{C}$ where $\tilde{K}(\mathbf{C}, \mathbf{S})$ could reach 1 have measure zero (this can be proved rigorously by outer regularity of the Haar measure $\mu$ on $H_d$). Thus, for $\mu$-almost every matrix, we could choose an $S$ with $\sum_{i=1}^d s_i^2 = d$, such that $||J_\phi(\mathbf{SCS})||_F < ||J_\phi(\mathbf{C})||_F$. That is, the scaling increases the stability of updates.

## H  CHARACTERIZATION OF THE OPERATOR IN THEOREM 1

For $\mathbf{T} \in DOS(u, v, w)$, its eigenspace has clear characterization, which is useful in the study of the behavior of $\mathbf{T}$. Let $\mathcal{M}_d$ be the subspace of $H_d$ spanned by those matrices only has off-diagonal

entries, which have dimension $\frac{d(d-1)}{2}$ and a basis $\mathbf{M}_{ij} = \mathbf{E}_{ij} + \mathbf{E}_{ji}$, where $\mathbf{E}_{ij}$ is the matrix with 1 on $(i, j)$ position and 0 anywhere else. And $\mathcal{L}_d$ is the span of $\mathbf{L}_i$, which is defined as:

$$
\mathbf{L}_i \triangleq \begin{pmatrix}
 & \cdots & & -v & & \cdots & \\
 & \cdots & & -v & & \cdots & \\
-v & -v & \cdots & w - u & -v & \cdots & \\
 & \cdots & & -v & \cdots & & \\
 & \cdots & & -v & \cdots & &
\end{pmatrix},
$$

where those non-zero entries are on $i-$th row and column.

**Theorem 4.** *For $\boldsymbol{T} \in DOS(u, v, w), w \neq u$, $\mathcal{M}_d, \mathcal{L}_d$ are its eigenspaces, with eigenvalues $w, u$ respectively.*

*Proof.* Here the condition $w \neq u$ ensures that $\mathcal{L}_d$ is linearly independent with elements in $\mathcal{M}_d$ since it spans the diagonal part of $H_d$. The results $\mathbf{T}\mathbf{M}_{ij} = w\mathbf{M}_{ij}, \mathbf{T}L_i = uL_i$ could be calculated using the definition equations of $DOS(u, v, w)$ and consider them on component level. Since $\mathbf{T}$ is a linear operator, verifying these eigen properties on the basis is enough for the result. Furthermore, the space we have specified spans a $\frac{d(d-1)}{2} + d = \frac{d(d+1)}{2} = dim H_d$ dimensional space, hence it is the whole $H_d$. Thus we have completely characterized the eigenspaces of such $\mathbf{T}$. $\qquad\square$

## I   DIAGRAM OF DRGCN

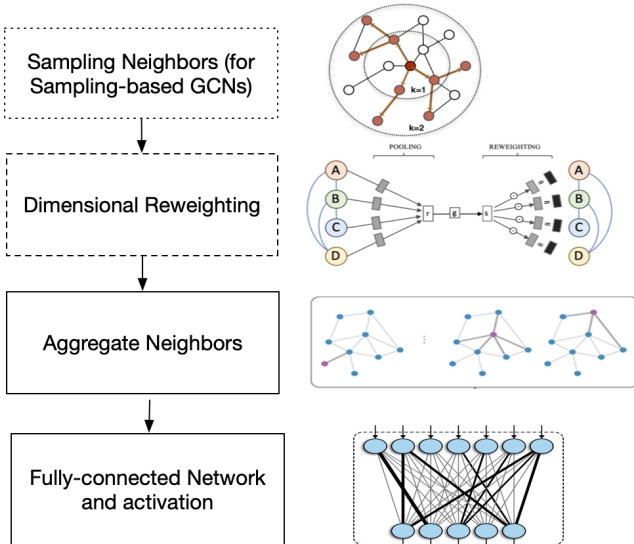

Figure 2: Diagram of a DrGCN layer. The "Sampling Neighbors" procudure is only applied in sampling-based DrGCNs.

## J   PRE-ACTIVATION GCNS

As discussed in section 3, Our theoretical analysis is based on the pre-activation setting, while common GCN methods use post activation. Although they are basically the same if we consider the activation in the pre-activation setting as the activation of the previous layer in the post-activation setting, there is still a little difference between pre and post activation that pre-activation activates the input feature. So we also experiment pre-activation on GCN, see table 13. Results are averaged among 20 runs.

Table 13: Results of GCN with pre and post activation on citation datasets.

| Method | CoraR | CiteseerR | Pubmed |
|---|---|---|---|
| GCN(pre activation) | 84.7± 0.7 | 74.1± 0.4 | 87.9± 0.3 |
| Dr-GCN(pre activation) | 85.1± 0.6 | 74.8±0.8 | 88.2 ± 0.2 |
| GCN | 85.9± 0.7 | 74.9 ± 0.7 | 88.0± 0.3 |
| Dr-GCN | 86.8± 0.5 | 77.5± 0.6 | 88.4± 0.3 |

