# OpenReview forum: "Dimensional Reweighting Graph Convolution Networks"
_ICLR.cc/2020/Conference — Reject_

### Official Review · AnonReviewer2 · 2019-10-26
**Official Blind Review #2**

**Rating:** 3

**Review:**

This paper proposes a method, known as DrGCN, for reweighting the different dimensions of the node representations in graph convolutional networks (GCN). Specifically, the representation of every node is element-wise multiplied with a weight vector, which is parameterized as a function of the average input node representation, where the function is a two-layer neural network.

At a conceptual level, this is similar to various existing normalization schemes, such as batch normalization and weight normalization. While it is claimed in Section 4.3 that the difference is that “[batch normalization] reduces variance between samples, while DrGCN reduces variance between dimensions”, I am not sure if this characterization is accurate. Batch normalization actually makes each dimension have unit (sample) variance and so does not make the variance of each dimension small. What it does is to make the sample variance of each dimension equal, which is also what DrGCN tries to do, i.e.: reduce variance across dimensions (since samples in the case of GCNs are the representations of different nodes). DrGCN is also similar to weight normalization because like weight normalization, DrGCN learns a transformation on top of the vanilla representation (in the case of weight normalization, the vanilla representation is the normalized weight vector; in the case of DrGCN, the vanilla representation is the node representation before reweighting). The conceptual contribution therefore seems incremental.

Incremental conceptual contributions would be fine if (1) they result in a surprising theoretical result, or if (2) they result in a surprising improvement in empirical performance. Unfortunately, neither seems to be demonstrated in this paper.

In the theoretical analysis, there are various occurrences of unjustified leaps of logic; as a result, what is claimed to be shown by the analysis is different from what is actually shown, and it is unclear what is actually shown is substantially related to the proposed method. For example, in Section 3.1, the paper says that “GCNs are different from fully-connected networks only in \tilde{A}, and degrade to fully-connected networks when \tilde{A} = I. So, our analysis can be somehow be generalized to DrGCNs.” The first sentence is true; in other words, it says that fully-connected networks are a special case of GCNs when \tilde{A} = I. However, the second sentence does not follow - just showing a special case (which is what the subsequent analysis does) does not say much about the general case. Relatedly, the architecture that is analyzed is of the form H^l = W^l \phi(S^l H^{l-1}) + b^l for l = 1, …, k, whereas the architecture that is proposed is H^l = \phi(W^l S^l H^{l-1} \tilde{A}^{l} + b^{l}) for l = 1, …, k. The latter cannot be cast as the former unless \tilde{A} is diagonal. Also, the caveat of the mean field approximation are not stated - whatever result that is shown is only valid at the infinite width limit, which is different from what is claimed in the abstract, which says that "We prove that DrGCNs can reduce the variance of the node representations by connecting our problem to the theory of the mean field.” Additionally, the analysis is done in the case where S is directly parameterized, whereas the proposed method parameterizes S as the output of a two-layer neural network. I presume the reason why latter was done in practice because it worked better empirically. Because this is not explained by the analysis, the theory is incomplete, and so this caveat should be clearly stated in the abstract.

In the experimental results, the performance improvement of DrGCN over layer normalization is not statistically significant. Also, DrGCN is only compared to other normalization schemes (batch normalization and layer normalization) on one dataset, and so there is insufficient evidence to conclude that DrGCN generally works better than existing methods empirically,

Additionally, in section 2.1, it is claimed that “sampling-based GCNs still lie within the framework of equation (2), as we can set all unsampled edges to 0 in \tilde{A}”. This does not seem to be accurate, because in sample-based GCNs, different edges are sampled in every minibatch, and so there is no fixed choice of \tilde{A} that makes these GCNs equivalent to the formulation in eq. (2).

Also, if the goal is to reduce variance across dimensions (as the paper claims in Section 4.3), why was the average node representation fed into the two-layer neural network rather than its reciprocal?

**Experience Assessment:**

I have read many papers in this area.

**Review Assessment: Checking Correctness Of Derivations And Theory:**

I carefully checked the derivations and theory.

**Review Assessment: Checking Correctness Of Experiments:**

I carefully checked the experiments.

**Review Assessment: Thoroughness In Paper Reading:**

I read the paper thoroughly.

---

> ### Author Response · Authors · 2019-11-08
> **Thanks for Review #2**
>
> Thanks for the comments.
>
> 1. About the "variance reduction" statement:
>
> Thanks for pointing out the inaccurate statement in section 4.3 and abstract, we have already fixed this in the revision. The proposed DrGCN method is not aimed to "reduce the variance between dimensions", instead, it's helpful by improving the training stability.
>
> 2. About the innovation points of DrGCN:
>
>  We propose K to measure the stability increment theoretically in section 3.2 to explain why our method may work, which is different from other feature normalization methods which only brings up a method with better performances.
>
> Previous methods are not using any normalization methods on Graph Convolutional Networks and the idea of adding an existing feature normalization method like batch-norm or layer-norm on top of the previous SOTAs can achieve improvements in performance on some datasets. However the performances of BN and LN are not quite stable and vary by methods and datasets. Further details of the results of batch-norm and layer-norm are included in Table 12 in the Appendix. On the contrary, the performance gain from DrGCN is more reliable with the support of the stability measure K.
>
> 3 About the completeness of the theory of DrGCN:
>
> For the theoretical part, it is true that our analysis is based on fully-connected networks. We follow the idea that analyses on fully-connected networks can provide insight into more generalized neural network structures. "A Mean Field Theory of Batch Normalization" (https://openreview.net/forum?id=SyMDXnCcF7) And our analysis does provide insight since the quantified measure K matches experimental results perfectly. We have made this clear in the revision.
>
> About the analysis of the matrix S and the constructed quantity K. Our idea is to construct a criterion of the stability of the update based on the result of learning. So the analysis is about quantifying the instability we have reduced through our matrix S instead of which matrix is directly parametrized. As mentioned in the review, the theory does not fully analysis the effectiveness of DrGCN. However, the theory on the stability measure K for training W is complete and provides very useful insight into the method. Though DrGCN's effectiveness is not fully explained by the stability measure K, it is rather more efficient experimentally when K is small.
>
> 4 About the construction of S:
>
> Our theoretical analysis provides an insight into the stability of training W, but S is also a parameter to be learned. Directly learning S leads to overfit and does not yield good results. Here our insight of the construction of S is from a widely adopted method in "Squeeze and Excitation Networks".(http://openaccess.thecvf.com/content_cvpr_2018/html/Hu_Squeeze-and-Excitation_Networks_CVPR_2018_paper.html) Though this paper doesn't provide any theoretical analysis or quantified measures, and its method does not directly work for GCNs, it does provide insight on how the construction of S may be good in practice, so we just modified its method to satisfy the GCN structure.
>
> 5 About $\tilde{A}$ for sampling GCNs:
>
> For sampling GCNs, yes they do not use a fixed $\tilde{A}$ on every step of the training process. However the framework does not require any of the weight matrices, including $\tilde{A}$, to be fixed. W also updates on every step during the training process. The point is that the framework and the proposed DrGCN works for sampling GCNs too.
>
> Thanks for the helpful comments. We have made a revision to fix all the mentioned inappropriate expressions and we have uploaded it .

---

### Official Review · AnonReviewer1 · 2019-11-04
**Official Blind Review #1**

**Rating:** 6

**Review:**

The paper is out of my research area. I could only provide little recommendation. I have tried to read this paper, but it was rather tedious with heavy notations. It would be more friendly to represent the models in visible way for example using diagrams as I can see that the model is a sequence matrix operators with non-linear transformations after that. The paper states that the proposed DrGCNs can improve the stability of GCN models via mean field theory. The experiments were conducted  on benchmark datasets and the proposed method was compared to several GCN variations.

**Experience Assessment:**

I do not know much about this area.

**Review Assessment: Checking Correctness Of Derivations And Theory:**

I did not assess the derivations or theory.

**Review Assessment: Checking Correctness Of Experiments:**

I did not assess the experiments.

**Review Assessment: Thoroughness In Paper Reading:**

I made a quick assessment of this paper.

---

> ### Author Response · Authors · 2019-11-08
> **Thanks for Review #1**
>
> Thanks for your positive feedback on our paper.
>
> 1 For the visible diagram of DrGCN
>
> Unfortunately we cannot attach figures in the reply. We have come up with a diagram in Appendix I in the revision.
>
> Thanks for the helpful comment.

---

### Official Review · AnonReviewer4 · 2019-11-04
**Official Blind Review #4**

**Rating:** 3

**Review:**

This paper proposes a method called "Dimensional Reweighting" for graph convolutional networks. The method involves a reparametrization of GCNs (by adding an extra reweighting block in each layer), which the authors show theoretically can reduce variance. The authors supplement this with extensive empirical experiments showing slightly improved performance by adding their method to existing methods.

I would vote to weakly reject this paper for two key reasons - first, I think the writing can be improved and explanations can be clarified, especially for people less familiar with the field like myself. Second, I am not certain how significant the final experimental improvements are (other than on the Reddit dataset), as most of the numbers are not that far apart, and it seems that different methods in the literature already produce fairly different results.

Overall, I think the structure of the paper is fairly good. I feel that a few things should be modified for clarity.
- You claim a 40% improvement in error rate in the intro, which sounds enormous. I would say "relative error rate" to avoid overclaiming, because 40% improvement sounds like you are reducing (absolute) error from 60% to 20%, while in reality you are reducing error from 3.6% to 2.1%.
- In section 2.1, did not know if the projection matrix W was learned or predefined.
- I was not sure why you used sigma_g and sigma_s as opposed to just sigma in equation (5). Do you use different activation functions? Also, I did not find what activation function the authors end up using in their experiments.
- I may have misunderstood something, but the theory does not seem to match the proposed method exactly. The mean-field theory analysis has the activation function after the reweighting by S but before multiplying by W, while the framework in Section 2.2 has the activation after the reweighting by S and after the multiplying by W. I am not sure how much this difference makes, or if it is significant, but I think it should be explained by the authors.
- I also did not understand exactly what the "variance" the authors are reducing is. The authors talks about "reducing the learning variance brought by perturbations on the input," but when is the input ever perturbed for GCNs? Explaining this more clearly would improve the motivation for this work.
- I would appreciate a better intuitive explanation of the measure "K." I gather that it is related to the "variance" being reduced, but it is different from that.

The experimental results are good overall, as the proposed method tends to give the best results (by a small margin) across the board. I especially appreciate that the authors performed many experiments over many different datasets and repeated runs 20 times to try to get confident estimates of how well each method performs. I also appreciate that the authors cleaned up the Citeseer and Cora datasets, and I hope the cleaned datasets will be useful for the research community.
With that said, I do not know how significant the improvement is. I think something that would be helpful would be to measure the "variance" that the method is supposed to be reducing (since it sounds like it is not exactly the same thing as K), and showing that in a table as well. This would show experimentally that the method achieves its intended goal.

Minor comments
- I would recommend that the authors proofread for English grammar and style in updated versions of the paper. For example, in the first paragraph of the introduction, the authors use "is proposed" instead of "were proposed" and typo "broad" as "board."
- Just curious, why did you choose a 2 layer network with 2 activation functions for the Dr block? Why not just have 1 hidden layer?

**Experience Assessment:**

I do not know much about this area.

**Review Assessment: Checking Correctness Of Derivations And Theory:**

I did not assess the derivations or theory.

**Review Assessment: Checking Correctness Of Experiments:**

I assessed the sensibility of the experiments.

**Review Assessment: Thoroughness In Paper Reading:**

I made a quick assessment of this paper.

---

> ### Author Response · Authors · 2019-11-08
> **Thanks For Review #4**
>
> Sorry for not being clear enough.
>
> 1 For the "error rate":
>
>  As pointed out, we do find that "reducing error rate by 40%" may be misunderstanding, we have corrected that to " number of misclassified cases reduced by 40%" in the revision.
>
> 2 For matrix W:
>
> In 2.1, the matrix $\mathbf{W}$ is learned, which is a basic concept for GCNs. See "Semi-supervised learning with Graph Convolutional Networks" (https://arxiv.org/abs/1609.02907)
>
> 3 For $\sigma_s$ and $\sigma_g$:
>
> $\sigma_s$ and $\sigma_g$ refer to different activations, detailed expressions are in the Appendix C.1.
>
> 4 For the difference in activation:
>
> We follow "A Mean Field Theory of Batch Normalization" (https://openreview.net/forum?id=SyMDXnCcF7) to use pre-activation as an approximate to post-activation. Pre-activation and post-activation does not do much difference on performance of GCNs. We experiment on post-activation just because previous methods use post-activation and we try to add Dimensional reweighting on top of them without changing their structures.
>
> The theory in our paper can also be applied to post-activation, since the S in the paper is a diagonal and positive matrix, and $\sigma$ is ReLU activation, thus $\sigma(SH)=S\sigma(H)$ , therefore $W\sigma(SH)+b=WS\sigma(H)+b$, which is the same as the post-activation format except for the activations on input features/output representations.  We have clarified this in the revision.
>
> 5 For the "variance":
>
> Sorry for that, this is just an inaccurate description, we have already fixed this in the revision. The proposed DrGCN method is not aimed to "reduce the variance between dimensions", instead, it is helpful by improving the training stability.
>
> 6 Intuiative Explanation of K:
>
> For the constructed quantity K, our idea is to construct a criterion of the stability of the update based on the result of learning. So the analysis of K is about quantifying the instability we have reduced through our reweighting scheme.
>
> 7 For the Construction of S:
>
> Our theoretical analysis provides an insight into the stability of training W, but S is also a parameter to be learned. Directly learning S leads to overfit and does not yield good results. Here our insight of the construction of S is from a widely adopted method in "Squeeze and Excitation Networks".(http://openaccess.thecvf.com/content_cvpr_2018/html/Hu_Squeeze-and-Excitation_Networks_CVPR_2018_paper.html) Though this paper doesn't provide any theoretical analysis or quantified measures, and its method does not directly work for GCNs, it does provide insight on how the construction of S may be good in practice, so we just modified its method to satisfy the GCN structure.
>
> Also $S$ generated this way is a diagonal positive matrix, therefore $S\sigma(H)=\sigma(SH)$
>
> Thanks for the helpful comments. We have made a revision to fix the mentioned inappropriate expressions and we have uploaded it.

---

> > ### Comment · AnonReviewer4 · 2019-11-12
> > **Clarification on pre-activation vs post-activation**
> >
> > Hi, thanks for your replies. Regarding (4), how do you know if the experimental results for GCNs are fairly similar between pre-activation vs post-activation? Has pre-activation also been done in previous works? And if so, could you also do pre-activation experiments as well, since this is the setting you actually do analyze theoretically? I realize I am asking a bit late in the revision process, so I will not hold the authors to doing this before the deadline, but this is a recommendation I would make for future updates.

---

> > > ### Author Response · Authors · 2019-11-13
> > > **A Quick Result on Pre-activation**
> > >
> > > Hi,
> > >
> > > We do a quick run for the pre-activation for GCN on coraR and citeseerR  (averaged among 20 runs and using the same hyperparameters as post-acv so maybe not fully-optimized.)
> > >
> > >                                 CoraR   CiteseerR
> > > Pre_acv                   84.7$\pm$0.7 74.1$\pm$0.4
> > > Pre_acv+Dr            85.1$\pm$0.6 74.8$\pm$0.8
> > > --------------------------------------------------
> > > Post-acv(in paper)85.9$\pm$0.5 74.9$\pm$0.7
> > > Post-acv+Dr           86.8$\pm$0.5  77.5$\pm$0.6
> > >
> > > It seems that pre_acv is a little worse than post_acv, and we haven't tuned the hyperparameters and just pick the post_acv hyperparameters (as described in appendix C in paper).

---

> > > > ### Comment · AnonReviewer4 · 2019-11-14
> > > > **Thank you for running these experiments**
> > > >
> > > > Thank you very much for running these experiments. I would suggest you include them in future revisions of your work.

---

> > > > > ### Author Response · Authors · 2019-11-15
> > > > > **Thanks for the suggestion.**
> > > > >
> > > > > We have included this in Appendix J in the revision.
> > > > >
> > > > > By the way, pre-activation can be viewed as post-activation of the previous layer. The only difference is that pre-activation does an activation on input features, which takes away some input information, so the result shall be a little worse.
> > > > >
> > > > > Therefore the theory in our paper can also be applied to post-activation, since the S in the paper is a diagonal and positive matrix, thus $\sigma(SH)=S\sigma(H)$, therefore $W\sigma(SH)+b=WS\sigma(H)+b$, which is the same as the post-activation format except for the activation on input features\output representations.
> > > > >
> > > > > Really thanks for your suggestions that inspired us to realize this.

---

### Decision · Program_Chairs · 2019-12-19

**Decision:**

Reject

**Comment:**

As Reviewer 2 pointed out in his/her response to the authors' rebuttal, this paper (at least in current state) has significant shortcomings that need to be addressed before this paper merits acceptance.